# Real-Time Humidity Measurement during Sports Activity using Optical Fibre Sensing

**DOI:** 10.3390/s20071904

**Published:** 2020-03-30

**Authors:** Chenyang He, Serhiy Korposh, Francisco Ulises Hernandez, Liangliang Liu, Ricardo Correia, Barrie R. Hayes-Gill, Stephen P. Morgan

**Affiliations:** 1Optics and Photonics Group, Faculty of Engineering, University of Nottingham, Nottingham NG7 2RD, UK; eexch9@nottingham.ac.uk (C.H.); ezzsk2@exmail.nottingham.ac.uk (S.K.); ezzll1@exmail.nottingham.ac.uk (L.L.); ezzrnc@exmail.nottingham.ac.uk (R.C.); eezbrhg@exmail.nottingham.ac.uk (B.R.H.-G.); 2Footfalls and Heartbeats UK Ltd, Innovation Building, Biocity, Nottingham NG1 1GF, UK; ulises@footfallsandheartbeats.com

**Keywords:** optical fibre sensor, humidity sensing, layer-by-layer, real-time measurement, smart textiles, wearable technology

## Abstract

An optical fibre sensor for monitoring relative humidity (RH) changes during exercise is demonstrated. The humidity sensor comprises a tip coating of poly (allylamine hydrochloride) (PAH)/silica nanoparticles (SiO_2_ NPs) deposited using the layer-by-layer technique. An uncoated fibre is employed to compensate for bending losses that are likely to occur during movement. A linear fit to the response of the sensing system to RH demonstrates a sensitivity of 3.02 mV/% (R^2^ = 0.96), hysteresis ± 1.17% RH when 11 bilayers of PAH/SiO_2_ NPs are coated on the tip of the fibre. The performance of two different textiles (100% cotton and 100% polyester) were tested in real-time relative humidity measurement for 10 healthy volunteers. The results demonstrate the moisture wicking properties of polyester in that the relative humidity dropped more rapidly after cessation of exercise compared to cotton. The approach has the potential to be used to monitor sports performance and by clothing developers for characterising different garment designs.

## 1. Introduction

Humidity, the amount of water vapour in the air, is an important indicator relating to human comfort and performance during work and sport. Tsutsumi et al. reported that low indoor humidity level has positive effects on human wellbeing, however, when the humidity reaches above 70% RH, subjects reported to be more tired [1]. In sports performance, Maughan et al. reported that the cycling time to exhaustion was reduced by 8.6 min, 14.5 min and 22.1 min when environment relative humidity (RH) levels were increased from 24% RH to 40%, 60% and 80% RH respectively [2]. Although human core body temperature is maintained in a comfort zone even during the work and sports activities due to the thermoregulatory system [3], the skin surface humidity has a strong influence on thermal comfort [4]. 

The relationship between comfort, performance and humidity has inevitably led to a wide range of different materials being developed by textile and clothing manufacturers [5]. ISO 15496:2018 standard describes a method for testing the water vapour permeability of textiles that are often adopted within the field. Although this is a useful first approach, performed in a controlled environment, the tests are static, conducted on fabric swatches and not on the fully assembled garment or in a real world environment. There is therefore interest in developing new technology to assess water vapour permeability of fully assembled garments on human subjects in climatic chambers or on the field of play. For example, Koptyug et al. demonstrated the moisture transport capacity of different sports garments through monitoring the RH from human body [6]. Electronic temperature and humidity sensors were positioned either side of garments developed for cross-country skiing. It was demonstrated that garments with incorporated microporous membranes had superior performance at low ambient air humidity but were less effective for moisture transfer from the body in the rain. Raccuglia et al. mapped sweat properties in different regions of a garment using infrared images after treadmill exercise [7]. Infrared images are assumed to be proportional to sweat adsorbed within the T-shirt after it has been removed. Salvo et al. developed an electronic wearable sweat rate sensor comprising a 3D printed Polytetrafluoroethylene(PTFE) chamber for sweat collection and read out electronics which monitored athletes’ hydration status during exercise [8]. A non-wearable medical device Dermalab^®^ trans-epidermal water loss (Cortex Technology, DK) was used as a reference. It is suggested that a distributed body network could be established to provide valuable information for improving athletes’ physical performance.

Compared to electronic humidity sensors, an optical fibre sensor (OFS) offers several advantages such as light weight, miniature size, flexibility, simple configuration and insensitivity to electromagnetic fields [9]. The diameter of a single mode optical fibre is typically 125 µm which is smaller than many textile yarns meaning the OFS can be embedded into garments for wearable sensing applications [9]. A range of different approaches for optical fibre humidity sensing have been proposed [10] including interferometers [11], evanescent wave interactions [12], in-fibre gratings (Bragg and long period gratings) [13,14,15], carbon nanotubes [16], photonic crystal fibres [17] and reflection measurements [18]. Most optical fibre humidity sensors require a hydrophilic material which can interact with water molecules, so that a property of the guided light (e.g., intensity or wavelength) changes with humidity levels. SiO_2_ nanoparticles based optical fibre sensors show fast response time and appropriate RH range for measurement [19].

Previous research has shown different biomedical applications of RH measurements such as skin microenvironment [20] and breath tests [19,21]. However, all these sensors involve static RH measurement that avoids the influence of macro-bending loss from optical fibre [22]. To the authors’ knowledge no papers have investigated the use of optical fibre humidity sensing during exercise. This can pave the way for new textile characterisation approaches and wearable devices. 

In this paper, an optical fibre sensing system is developed and applied for RH measurement during sports activity (in this case a cycling test). Each sensor consists of two optical fibres. One with Poly(allylamine hydrochloride) (PAH, a highly hydrophilic polycation layer)/SiO_2_ NPs thin film for measurement of relative humidity and the other is a bare fibre used as a reference for bending losses compensation during exercise. Different tee-shirts are investigated: 100% cotton where the material will adsorb moisture and 100% polyester which has well known moisture wicking properties. These were selected to demonstrate that the properties of the fabric can affect humidity rather than to demonstrate the well-established differences between a natural and a synthetic garment.

## 2. Materials and Methods

### 2.1. Materials

Potassium hydroxide (KOH), Sodium hydroxide (NaOH), Ethanol and Poly (allylamine hydrochloride) (Mw ~58,000, PAH) were purchased from Sigma-Aldrich, UK. Silica Nanoparticles (SiO_2_ NPs, diameter 40–50nm) were purchased from Nissan Chemical, Japan. Deionised water (DI water), having resistivity of 18.2 megohm, was obtained from a water purification system (PURELAB Option S/R, ELGA).

### 2.2. Apparatus

Figure 1 shows the sensing system including the designed probe. The system consists of a Superluminescent Diode (SLD) (Superlum, SLD-761-HP, UK) as a light source with central wavelength 1550 nm and a full width at half maximum (FWHM) bandwidth of 45 nm. The photodiode (RS, FCI-InGaAs-120L-FC 900 nm–1700 nm, UK) is connected to a transimpedance amplifier (TIA) (LM358, Texas Instruments) to convert the light signal to a voltage output. An optical switch (Sercalo Microtechnology, SW1*4-9N, Switzerland) containing 4 channels is used to receive the reflected light from two sensors. A micro-controller (Arduino, Uno) is utilized to acquire signals (sampling frequency 1 Hz, 16 Bit resolution) and operate the optical switch. Light emitted from the SLD transmits through a 1*2 50/50 optical fibre coupler (All4fiber, SPL9-12-5050-NC-FP) to a port of the optical switch. 

The sensing probe is composed of two optical fibres (Corning, SMF-28e+, fibre diameter: 242 ± 5 µm, UK) which are an optical fibre humidity sensor (OFHS) with PAH/SiO_2_ NPs coating and a reference fibre (REF) without any coating. The end of each fibre was perpendicular cleaved. Two fibres are tightly surrounded by a 900 µm furcation tube (Thorlabs, FT900Y, UK) and heat shrink tubing (TE Connectivity Raychem Cable Protection, X4-0.8-0-SP-01-ND, recovered inner ϕ: 400 µm, UK) for protecting the tip from mechanical damage. Two pairs of sensors are used in this study.

### 2.3. Sensing Probe Fabrication & Relative Humidity Sensing Mechanism

A thin film composed of PAH/SiO_2_ NPs was coated on the tip of an optical fibre using the Layer-by-Layer self-assembly method [23]. The fabrication method is as follows: (a) the surface of the fibre tip was immersed into a KOH solution (1 wt % in ethanol/water = 3:2, v/v) for 25 min to hydroxylate the fibre; (b) after washing with DI water and drying with nitrogen, the fibre was treated with a 0.17 wt % positively charged PAH solution (pH: 10–11) for 15 min; (c) wash and dry, the fibre was then immersed into a 5 wt % negatively charged SiO_2_ NPs solution for 15 min. (b) and (c) processes were repeated to achieve the required thickness. The coated optical fibre was dried at room temperature in the laboratory for 24 h before use. For the layer by layer technique, the thickness of each layer is approximately equal to the diameter of SiO_2_ NPs i.e., a 40–50 nm/monolayer. To determine the optimum film thickness, sensors with 3, 7 and 11 layers were prepared. 

The humidity sensing mechanism of a PAH/SiO_2_ NPs coated optical fibre sensor has been described previously [19]. The total reflection intensity due to Fabry-Perot cavity created by functional film deposition can be explained as follows: (1) the optical fibre sensor behaves as a Fabry-Perot interferometer of low finesse, two main reflections were generated by two optical interfaces (fibre-film and film-air); (2) reflected light changes are due to changing the optical thickness of the film; (3) losses are due to light scattering and roughness of the film surface. The total optical response can be modulated by adsorption and absorption of water molecules by means of changing the effective refractive index of the film. The effective refractive index of the PAH/SiO_2_ NPs is approximately 1.22 and the intensity changes linearly with RH [19].

Meanwhile, water vapour does not easily adhere to the smooth tip surface of the uncoated bare fibre, which means that the bare fibre with perpendicular cleaved tip has the potential to be a reference for reducing bending loss effect. Ideally, under the same bending conditions, the intensity-loss coefficients (*α*) between two separate optical fibres from one batch are identical [24]. 

In this work, the intensity change due to bending of the OFHS is compensated using a reference fibre. We define a term *α* as a loss of intensity due to an arbitrary bending angle *θ* normalised by the intensity at a bending angle *θ* = 0.
(1)$$\alpha =\frac{\Delta I_{ref}}{I_{ref\;\left(\theta =0\right)}}$$
where Δ*I_ref_* is the change in intensity between the reference fibre at *θ* = 0 and at a bending angle *θ*. Δ*I_ref_*(*θ* = 0) is the intensity measured from the reference fibre at *θ* = 0 which can be made before measurements commence. As the (unknown) bending angle *θ* is the same in both reference and sensing fibres, *α* is then used to compensate the humidity sensing channel.
(2)$$I_{compensated}=I_{OFS}*\left(1-\alpha \right)$$
where *I_OFS_* is the intensity measured from the sensing optical fibre. 

### 2.4. Microscopy

The fibre before and after coating with PAH/SiO_2_ NPs was imaged using a transmission optical microscope (Olympus, BX50, UK). The morphology of the functionalised fibre tip was studied using scanning electron microscopy (SEM) (JEOL, JSM-7100F, UK). 

### 2.5. Calibration and Evaluation of Humidity Sensing System 

A climatic chamber (CVMS Climatic, Bench Top, RH range: 20% RH - 98% RH, humidity fluctuation: ±2.5% RH, temperature range: −20–180 °C) was used in the calibration process. According to the temperature humidity control range, Appendix A, the humidity in the climatic chamber should be set in the range from 45 to 98% RH at ~30 °C. The sensing probe and a commercial capacitive humidity sensor (Bosch, SparkFun BME280 atmospheric sensor breakout, RH range: 0% RH–100% RH; accuracy tolerance: ±3% RH; hysteresis: ± 1% RH and response time: 1s in the temperature range of −40 °C to 85 °C) were placed 0.5 cm apart inside the climatic chamber. 

The sensor response has previously modelled as linear [19,20] and so the RH was changed from 50% RH to 85% RH and reversed every 10 min with three cycles to obtain the line of best fit. In order to determine the highest sensitivity to RH, functionalised fibres with different numbers of PAH/SiO_2_ NPs bilayers were tested.

For verification, the sensing probe and commercial humidity sensor were placed 0.5 cm apart inside the climatic chamber. Using the calibration curve obtained from RH calibration experiment, the output signal from the sensing probe was confirmed in terms of RH unit. Thus, the RH in the climatic chamber was set from 55% RH to 95% RH and reversed with two cycles. 

### 2.6. Humidity Measurements during Exercise

In order to observe the performance in sports activities of the optical humidity sensing system, a cycling experiment was performed. 10 volunteers (male, weight: 61–90 kg, median weight: 71.5 kg, age 23–48, median age: 33) participated in the cycling test using an indoor cycle (Domyos, VM590, UK) at room temperature (22 ± 1 °C). Each volunteer wore two sports T-shirts with different materials (100% cotton and 100% polyester) separately for the same cycling test. The two cycling tests were separated by 45 min in order to stabilise the physical condition of volunteers. All the T-shirts were laundered and conditioned in the lab for 24 h before testing. Ethical approval was provided by the Ethics Committee, Faculty of Engineering, University of Nottingham. 

In order to observe the performance in sports activities of the optical humidity sensing system, a cycling experiment was performed. 10 volunteers (male, weight: 61–90 kg, median weight: 71.5 kg, age 23–48, median age: 33) participated in the cycling test using an indoor cycle (Domyos, VM590, UK) at room temperature (22 ± 1 °C). Each volunteer wore two sports T-shirts with different materials (100% cotton and 100% polyester) separately for the same cycling test. The two cycling tests were separated by 45 min in order to stabilise the physical condition of volunteers. All the T-shirts were laundered and conditioned in the lab for 24 h before testing. Ethical approval was provided by the Ethics Committee, Faculty of Engineering, University of Nottingham. 

The commercial humidity sensor BME280 was also used for verification of the signal from the optical fibre humidity sensor. BME280 and OFHS were embedded in a 3D-printed holder (Top left inlet of Figure 2) for maintaining the same sensing direction. Two sensing units were attached in the back area of the sports T-shirt (inside and outside within 0.5 cm distance from each other) by double-sided tape. The cycling protocol was the following: Pre-condition by sitting for 10 min; pedal for 5 min at velocity of 3.3–3.5 km/h; static position for 10 min. 

## 3. Results

### 3.1. Optical Fibre Sensor Modification 

Sensors of with 3, 7 and 11 layers were fabricated. As an example of the coated tip, Figure 3a shows the cleaved fibre before and after coating with 11 layers of PAH/SiO_2_ NPs film and demonstrates that a uniform thin film was coated on the tip of the optical fibre. The SEM image (Figure 3b) shows the porous morphology of PAH/SiO_2_ NPs film deposited onto the fibre.

### 3.2. Sensor Calibration.

Figure 4a shows the dynamic changes in the reflection light intensity of the OFHS modified with different bilayers (3, 7 and 11) of PAH/SiO_2_ film upon exposure to the changing of RH in the climatic chamber. It should be noted that the OFHS response (blue curve) is broadly repeatable and inversely proportional to the RH change measured by the commercial humidity sensor (black curve). Comparing the dynamic light changes of OFHS with different layers (3, 7 and 11), Figure 4a shows the increase in the geometrical thickness of the porous morphological film provides a higher sensitivity as more water vapour can be adsorbed. However, thicker films provide less reflected light [19,25], which can be observed in Figure 4a. Therefore, to obtain a balance between sensitivity and detected signal level 11 bilayers are utilised in the remaining experiments.

Figure 4b shows the calibration curves for RH measurements with OFHS with 3, 7 and 11 layers of PAH/SiO_2_ thin film. For all sensors, the light intensity change was observed to exhibit an approximately linear dependence on the change of RH (50–80%). It can be observed that the greater the number of layers leads to an increase in the sensitivity of the OFHS. For the sensor with 3 layers (red line), the sensitivity was 0.40 mV/% RH (R^2^ = 0.95), for 7 layers (dark blue line), the sensitivity was 0.96 mV/% RH (R^2^ = 0.88) and for 11 layers (orange line) the sensitivity increased to 3.02 mV/% RH (R^2^ = 0.96). The hysteresis value is ±1.17% RH for 11-layers humidity sensor. The hysteresis is defined as the difference between measurements of the humidity increase/decrease branch and the averaged curve of both branches, the sequence 50-85-50% RH is used as shown in the Appendix A. 

### 3.3. RH Measurements 

#### 3.3.1. Bland-Altman Analysis of RH Measurement

The RH results obtained from the developed OFHS with 11 layers and commercial sensor can be compared in Figure 5. As shown in Figure 5a, the two RH readings provide a similar trace during the two cycles of changing RH (from 60% RH to 92% RH). Using Bland-Altman analysis [26], Figure 5b demonstrates that the difference between the two sensors is similar in the RH range from 60% to 92%, although some excess scatter can be seen at 70–80% RH. As can be seen in Figure 5b, the mean and standard deviation of the difference is 1.44% RH and 1.41% RH, respectively.

#### 3.3.2. Bending Test 

In order to investigate the bending response of the sensor, the 11 layer sensing probe was further tested using the set-up shown in Figure 1 and bending the part of the sensing probe with shrink tubing (between the tip and the coupler) over a range of bend radius (up to 5 mm) to simulate typical movements of the fibre that might occur during exercise.

Figure 6a demonstrates the response of the OFHS and REF fibres when bending with different bend radius in parallel. The intensities show good consistency under the bending action over the radius range. Figure 6b shows the output of sensing probe before (red curve) and after (dark blue curve) compensation from the reference fibre during the random bending test. It is clear that the system is robust to motion artefacts after bending loss compensation was carried out. 

#### 3.3.3. RH Measurements during Exercise

Figure 7 shows one of the results of RH measurement from the OFHS and commercial sensor during cycling exercise whilst riding an indoor bike. The traces of the OFHS show a similar trend to the commercial device, although there was a slight absolute offset difference between the RH reading of the OFHS and commercial sensor. This is likely caused by the measurement position of the OFHS and commercial devices being slightly different, therefore the RH readings from the two sensors represent slightly different values. 

Cotton is a hydrophilic material and polyester has hydrophobic property [7,27]. It can be seen from the two graphs that there is a significant difference in RH readings between cotton and polyester cases. The amplitude of the RH reading has reached a maximum of 73.3% (OFHS) in the cotton case and a maximum of 47.7% in polyester. Moreover, when the volunteer wears a polyester T-shirt the humidity levels recovered much quicker than that in cotton case when the subject rests.

Table 1 shows the full width at half maximum (FWHM) of RH measurement curve obtained inside the garment and the difference between the RH peak value (Δ_peak_) of inner and outer sensors from the OFHS for volunteers wearing 100% cotton and 100% polyester T-shirts. As can be seen from Table 1, the 100% polyester group has a significantly lower FWHM value than the 100% cotton group, which means the generated humidity is released more quickly in the polyester case. In addition, the majority of Δ_peak_ readings with polyester were smaller than that in the cotton group. This confirms the performance of polyester as a moisture wicking material [28].

## 4. Discussion

The developed sensing system has enabled relative humidity to be measured during exercise. Due their flexibility and small diameter (smaller than most textile yarns), optical fibre sensors can be easily integrated within textiles with the potential to form next generation smart garments. They are more unobtrusive than commercial electronic humidity sensors but, for future widespread use, need to be competitive on cost. The prototype demonstrated in this paper comprises several components such as an SLD, fibre optic coupler and optical switch that would limit its use to a laboratory-based testing tool. However, for widespread use in wearable sensing the system could be simplified and reduced in cost. For example, multiple LEDs and photodiodes could be used to replace the SLD and optical switch. The bill of materials for a single channel system with wireless electronics is estimated to be ~€25 with the most significant cost a wireless microcontroller. As with most electronics systems, this cost would reduce with higher volumes.

The system is robust to motion artefacts due to the inclusion of a reference fibre that can be used to compensate for bending losses as shown in Figure 6b where the sensing and reference fibre are bent in random orientations. There is a small residual signal in the compensated channel in Figure 6b as there is a slightly different response in the fibres. This is due to light travelling in slightly different paths depending on the bend orientation of the fibre. Appendix A shows an example of 2 bare fibres bent at different orientations, angles and with different bend radii. The worst case difference in intensity over a wide range of bend angles and radii is 8% with the majority of cases less than 3%. The reference fibre is uncoated and so should be unaffected by humidity. Although not observed in these experiments, there is potential for the reference signals to be affected by condensation at the fibre tip and so a future design would benefit from a hydrophobic coating on the reference sensor. 

In line with previous research [19,29] a linear fit through the intensity versus relative humidity data has been used to determine the sensitivity with 11 layers demonstrating the highest sensitivity of 3.02 mV/% (R^2^ = 0.96). Figure 6 shows good agreement with a commercial sensor, however, this may be improved further by investigating non-linear fitting. 

The measurement unit is currently a bench top system with a trailing optical fibre between it and the subject. In certain applications e.g., climatic chamber or treadmill exercise, this is not a significant problem, although bending losses will increase due to the long optical fibre cable between subject and measurement unit. However, for measurements on the field of play or workplace, a wearable unit with wireless communications is desirable. The instrumentation utilised here is relatively simple and compact, and so miniaturising the measurement unit will be a focus of future work. The system currently measures RH on the inside and outside of a garment at a single position but due to the small size of the sensors there is potential to increase this by increasing the number of sensors and number of channels on the optical switch, for example, 16 is likely to be deployed in the next version of the device. 

The system has been tested on two garments with well-known and different properties (100% cotton and 100% polyester). The aim was to demonstrate the potential of the sensor and not to characterise the materials. Future tests will investigate different materials, structures and multiple garment layers.

## 5. Conclusions

A new approach to monitoring humidity in garments based on optical fibre sensing has been demonstrated. An optical fibre coated with an eleven-layer PAH/SiO_2_ nanoparticle film was used to measure changes in light intensity reflected from the tip of the fibre which are proportional to humidity. Bending losses were partially compensated by including an uncoated reference fibre which enables measurements to be made during exercise. 

The response of the sensing system to RH concentration was linear with a sensitivity of 3.02 mV/% (R^2^ = 0.96) when 11 bilayers of PAH/SiO_2_ NPs coated on the tip of fibre was used which was higher than that of 7 (0.96 mV/%, R^2^ = 0.84) and 3 bilayers (0.40 mV/%, R^2^ = 0.93) of PAH/SiO_2_ nanoparticles. Measurements from 10 volunteers during a cycling test demonstrate the capability of the system to measure the effect of textile properties on humidity, with polyester providing better moisture wicking properties than cotton. The small size and flexibility of the sensors allow them to be easily integrated within garments with negligible effect on the moisture wicking properties and has the potential to be used as a tool for clothing developers designing moisture managing products. 

## Figures and Tables

**Figure 1 sensors-20-01904-f001:**
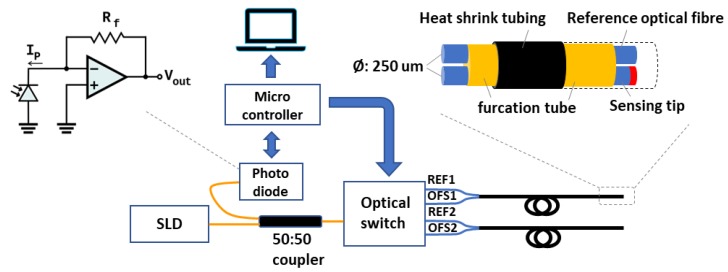
Humidity sensing system. Light is emitted from the light source and through an optical fibre coupler transmits to a port of the optical switch. The photodiode and transimpedance amplifier convert the reflected light intensity to a voltage which is then transferred to the pc via the microcontroller.

**Figure 2 sensors-20-01904-f002:**
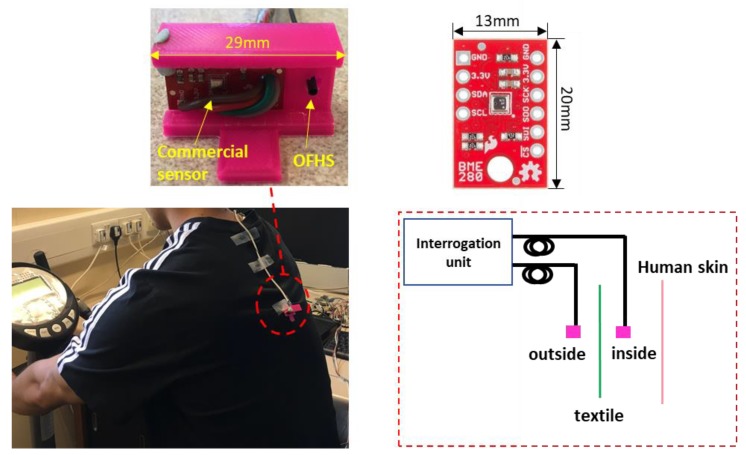
Humidity measurement in sports activity (cycling). Two pairs of sensors (BME280 and OFHS) were placed inside and outside of the garment respectively. BME280 and OFHS were embedded in a small 3D-printed holder for maintaining the same sensing direction.

**Figure 3 sensors-20-01904-f003:**
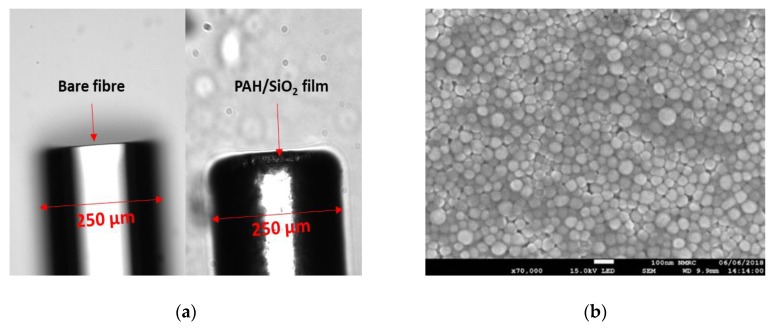
(**a**) Optical microscope images of cleaved fibre before (left) and after coating with 11 layers of PAH/SiO_2_ NPs (right) (**b**) SEM image of PAH/SiO_2_ NPs film deposited onto a fibre tip. The value of the scale bar is 100 nm.

**Figure 4 sensors-20-01904-f004:**
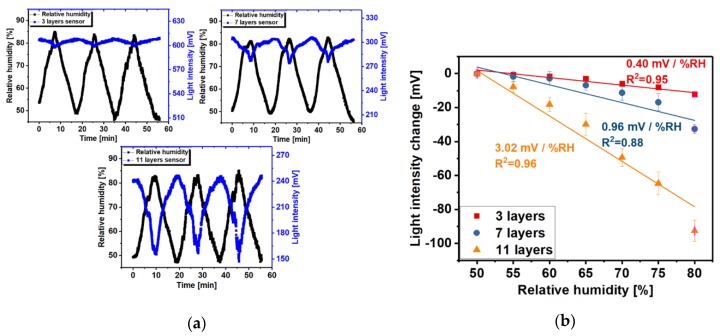
(**a**) Dynamic light intensity changes of OFHS with different layers (top left: 3 layers, top right: 7 layers, bottom: 11 layers) thin film (blue curve, PAH/SiO_2_ NPs) on exposure to a humidity changing environment. The black curve is the RH change measured by the commercial sensor. The temperature is fixed at 30 °C in the climatic chamber for the duration of the experiment. (**b**) Calibration curves for OFHSs with 3, 7 and 11 layers thin film represent the sensitivity for each case (error bars represent the standard deviation of three repeat cycles). For the purpose of comparison, the Y-axis in the calibration curve shows the differences in reflected light intensity relative to the left experimental value.

**Figure 5 sensors-20-01904-f005:**
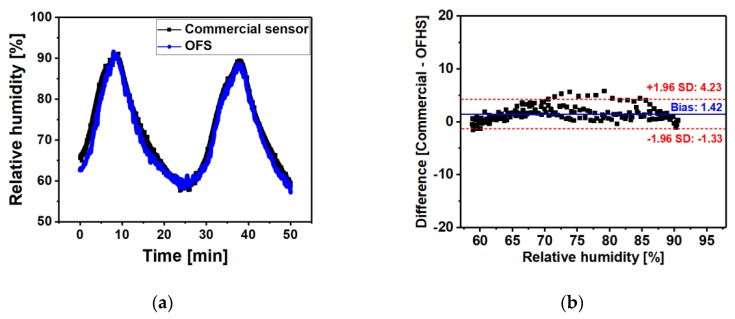
(**a**) The RH outputs from the 11 layer sensing probe (blue curve) and commercial sensor (black curve) measured in the climatic chamber. (**b**) Bland-Altman plot showing the difference between RH measurements from sensing probe and commercial humidity sensor. The x-axis and y-axis show the relative humidity and the difference (Commercial - OFHS) of the RH reading from the two sensors, respectively. The red dashed lines represent the 95% limits of agreement = 1.96SD of the differences between two sensors.

**Figure 6 sensors-20-01904-f006:**
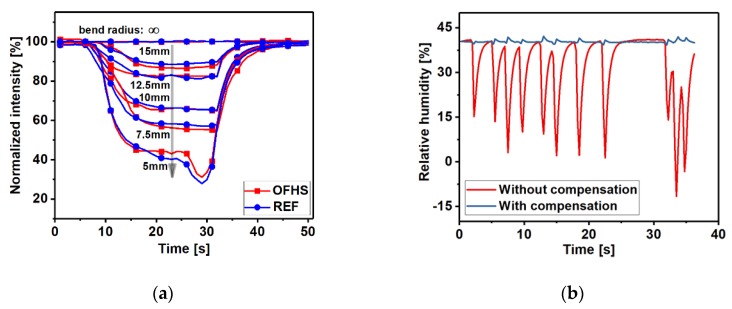
(**a**) The response from OFHS and REF fibre during bending over a range of bend radius (up to 5 mm). (**b**) The output from the sensing probe during random bending. The red trace is the RH output without compensation from the reference fibre and the dark blue trace is after compensation.

**Figure 7 sensors-20-01904-f007:**
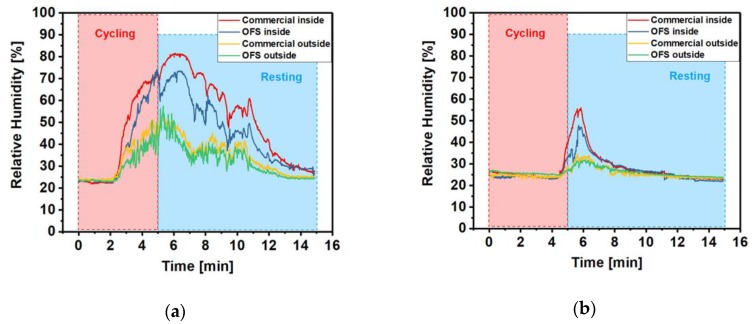
The RH reading from the sensing probe (inner, blue curve and outer, green curve) and commercial sensor (inner, red curve and outer, yellow curve) measured during the cycling exercise with 100% cotton T-shirt (**a**) and 100% polyester T-shirt (**b**) of volunteer 1.

**Table 1 sensors-20-01904-t001:** Full width at half maximum (FWHM) and Δ_peak_ of the OFHS RH reading under different textile (100% cotton and 100% polyester) for 10 volunteers.

Volunteer Number	FWHM (s)		Δ_peak_ (RH%)	
	Cotton	Polyester	Cotton	Polyester
V1	313.8	75.6	23.72	16.23
V2	296.4	181.8	38.38	26.69
V3	354	148.8	28.68	51.75
V4	247.2	168.6	24.01	36.24
V5	580.8	384	30.39	26.56
V6	484.2	334.8	37.66	29.29
V7	577.2	258.6	28.66	21.44
V8	*	407.4	41.42	38.35
V9	314.4	309.6	17.72	9.61
V10	247.8	69.6	30	10.93

* In the V8 cotton case, the RH did not recover to the half maximum before the end of the experiment due to profuse sweating.

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
