# Peer review of "Real-Time Humidity Measurement during Sports Activity using Optical Fibre Sensing"

_sensors, 2020, doi:10.3390/s20071904_

Round 1

Reviewer 1 Report

Although the fiber sensors for humidity are not novel, the application to test the garment's humidity is interesting. However, the paper was not written standardly and there are some flaws in the experiments.

  1. In lines of 130 to 136, how is the ΔIref defined? How to measure the Iref? To compensate the bending loss, the functions (3) and (4) are not applicable. you should give the algorithm in the demodulation.
  2. In section 3.1, why did the authors choose 11 layers for the sensors? In Fig. 5, the sensing responses of sensors with 3 or 7 layers are also given. These text should not be at different places. An overall illustration for the configuration of sensors should be given. As for Fig. 5, the data are linear fitted. What is the linearity? The sensors with 7 layers have a high sensitivity but a low linearity.
  3. The commercial sesor performed as a calibrated sensor to test the sensor in this paper, whose datasheet should be presented, such as the linearity, sensitivity, precision.
  4. In fig. 4, it is difficult to find the symmetric and linearity.
  5. Hysteresis is very important for humidity sensor and should be illustrated in the paper.

Reviewer 2 Report

The authors present a work of fabrication and characterization of a fiber-optic humidity sensor. The work is mostly focused on a specific application, which is humidity monitoring to evaluate textiles for sports clothing. In particular, the sensor was attached to the back of a cyclist, one in the inner and one in the outer side of the garment. Two textiles were compared, cotton and polyester, showing very different humidity values. The tests are verified with an independent commercial humidity sensor to validate the results.

In general, the paper is well presented, scientifically sound, and shows an actual application of an optical fiber sensor. I think the paper deserves publication; however, there are a few points that need clarifications:

1) The authors strongly focus the paper on the application. In this case, the importance of sensor cost is paramount. However, the authors do not address this issue. It would be useful to estimate how low the cost can become after a hypothetical industrialization of the sensor. Current electronic sensors are very inexpensive, so it will be difficult for a fiber sensor to compete in this field. The authors should dedicate at least one paragraph to this aspect.

2) In line 112, the concentration of the KOH solution is not indicated (only the ethanol-water ratio, but not the KOH concentration).

3) In Eq. 1 they explain the transducing mechanism. But that equation is only valid for semi-infinite media. When the layer is thin, the response may show fringes, so the authors should indicate the thickness of the deposited layer in order to understand if the equation is valid. What is the refractive index of the porous layer? I presume that the porous medium behaves as a homogeneous optical medium due to the small pore sizes. However, when humidity condenses in the pores, this approximation may not be valid anymore, so some clarifications would be useful.

4) In Fig. 5 the authors compare the responsivities of probes with different numbers of layers. However, only the relative signal variations with respect to humidity are shown. In the text, they say that the absolute readings are lower in thicker probes, so there is trade-off between sensitivity and signal. But they do not show this data in any figure. They should show the data of the absolute signal to understand how low the signal becomes.

5) Why did the authors exclude relative humidities below 50% in Figs. 4 and 5? If the reason is that the response became nonlinear, they still should include the data, rather than hide it.

6) To compensate for bend radius loss, the authors included a reference probe with no sensitive layer at the tip. This is crucial in any intensity-based optical sensor. However, when dealing with bend loss, the cable shape is very important. The fibers are in parallel, so when the bend is along the plane of the fibers, the bend is the same, but when the bend is in the perpendicular plane, the bend loss may be very different in both fibers, affecting the compensation algorithm. The authors should describe how the bending was applied during the tests.

7) Humidity response usually show hysteresis associated to the condensation and evaporation. Did the authors quantify this effect?

Round 2

Reviewer 1 Report

The paper has been improved greatly. But some equations do not display properly, such as eq. (1) and (2).
